# First Symptomatic Pediatric Case of Hb Rothschild (*HBB*: c.112T>C, p.Trp38Arg): Low-Oxygen-Affinity Hemoglobin Presenting with Persistent Pseudohypoxemia

**DOI:** 10.3390/diagnostics15243181

**Published:** 2025-12-12

**Authors:** Ekaterina Nuzhnaya, Andrey Marakhonov, Artem Ivanov, Yulia Lashkova, Ivan Kuznetsov, Tatiana Kulichenko, Ksenya Zabudskaya, Oxana Ryzhkova, Nikolay Zernov, Natalia Semenova

**Affiliations:** 1Federal State Budgetary Institution, Research Centre for Medical Genetics, 1 Moskvorechye Street, Moscow 115522, Russiasemenova@med-gen.ru (N.S.); 2Russian Children’s Clinical Hospital, a Branch of Pirogov Russian National Research Medical University, 117 Leninsky Prospekt, Moscow 119571, Russia; 3Biotech Campus LLC, Moscow 117997, Russia

**Keywords:** low-oxygen-affinity hemoglobinopathy, Hb Rothschild, pulse oximetry, pseudohypoxemia

## Abstract

**Background**: Hemoglobin Rothschild (Hb Rothschild), NM_000518.5(*HBB*):c.112T>C, is an ultra-rare low-oxygen-affinity hemoglobin variant that persistently causes reduced peripheral oxygen saturation on pulse oximetry despite normal arterial oxygenation. Fewer than ten cases have been reported worldwide, and only one involved a child—an asymptomatic carrier identified incidentally. **Methods**: The patient underwent clinical examination, growth assessment, blood tests, hemoglobin electrophoresis, chest CT, abdominal ultrasound, echocardiography, and pulmonary perfusion scintigraphy. Whole genome sequencing (WGS) of the proband and parents was performed, followed by bioinformatic analysis and ACMG-based variant interpretation. A PRISMA-guided PubMed literature review was conducted. **Results**: We report on the first pediatric case exhibiting a symptomatic clinical course. A 4-year-old boy was referred for chronically low peripheral oxygen saturation (SpO_2_), 78–86%, on pulse oximetry and recurrent lower respiratory tract infections. Early developmental history revealed episodes of apnea in infancy, perioral cyanosis, poor exercise tolerance, and low weight gain. Repeated cardiopulmonary assessments, chest computed tomography (CT), echocardiography, and pulmonary perfusion scintigraphy yielded unremarkable findings. Arterial blood gas analysis consistently showed normal arterial partial pressure of oxygen (PaO_2_), excluding true hypoxemia. Hemoglobin electrophoresis revealed an abnormal HbD fraction; WGS identified a heterozygous variant NM_000518.5(*HBB*):c.112T>C inherited from the patient’s asymptomatic father. This variant increases the partial pressure of oxygen at which hemoglobin is 50% saturated (p50), thereby decreasing hemoglobin’s oxygen affinity and shifting the oxyhemoglobin dissociation curve to the right. These alterations explain the discordance between low peripheral oxygen saturation (SpO_2_) and preserved oxygen delivery to tissues. **Conclusions**: This case expands the clinical spectrum of Hb Rothschild and demonstrates that symptomatic presentation may occur in early childhood. Awareness of low-affinity hemoglobin variants is essential to avoid misdiagnosis and unnecessary cardiopulmonary interventions. Early genetic testing facilitates accurate diagnosis and appropriate counseling.

## 1. Introduction

Hemoglobin (Hb) is a tetrameric protein composed of two α-like and two β-like globin chains, each containing a heme group responsible for reversible oxygen binding. The genes encoding α-globin are located on chromosome 16, whereas the β-globin genes, including *HBB*, are located on chromosome 11 and are expressed after birth. Structurally, Hb exists in two quaternary conformations: the deoxygenated tense state (T-state) and the oxygenated relaxed state (R-state). The transition between these states underlies the cooperative binding of oxygen [1,2,3].

Hemoglobinopathies caused by structural variants of globin chains may alter hemoglobin’s affinity for oxygen. Variants with low oxygen affinity shift the oxygen dissociation curve to the right and may produce falsely low peripheral oxygen saturation (SpO_2_) despite preserved arterial partial pressure of oxygen (PaO_2_). This phenomenon—pseudohypoxemia—often leads to misinterpretation as cardiopulmonary disease and results in unnecessary diagnostic interventions [4].

Hb Rothschild is an exceptionally rare structural hemoglobin variant that alters hemoglobin–oxygen binding dynamics, resulting in persistently low SpO2 unrelated to cardiopulmonary disease. The literature describes several adult cases, most of them detected incidentally and without clinical symptoms or functional limitations [5,6,7,8,9]. Only one pediatric case has been previously reported, characterized by asymptomatic desaturation discovered during routine examination [10].

Here, we present a pediatric case of Hb Rothschild with pronounced clinical manifestations that appeared within the first years of life and was inherited from an asymptomatic father. This report expands the current understanding of the clinical spectrum of this rare hemoglobin variant and emphasizes the importance of recognizing low-oxygen-affinity hemoglobinopathies in children with unexplained low oxygen saturation.

## 2. Case Presentation

### 2.1. Clinical Evaluation

The proband was a 4-year-old male. He was born to non-consanguineous parents at 39 weeks of gestation after an uneventful pregnancy and spontaneous vaginal delivery. His birth weight was 2750 g, and his length was 50 cm. The Apgar scores were 8 and 9 at 1 and 5 min, respectively. The perinatal period was unremarkable, and he was discharged home at the appropriate time.

Early developmental milestones showed a slight delay in gross motor development, with independent walking achieved at 15 months, whereas psychomotor and speech development were otherwise within normal limits. During the first year of life, episodes of apnea accompanied by decreased oxygen saturation were reported. After the onset of independent walking, the parents also noted weakness in the lower limbs during ambulation. From early infancy, nasolabial cyanosis was consistently observed, and the child demonstrated increased fatigability during physical activity, which limited his participation in age-appropriate motor and play activities.

From the age of 2 years and 2 months, the parents observed that episodes of upper respiratory infections became more severe. During this period, an episode of febrile illness was accompanied by documented oxygen desaturation to 86%, which required emergency hospitalization. SARS-CoV-2 infection was excluded. Chest CT was unremarkable, whereas abdominal ultrasound revealed transient signs of partial bowel obstruction. According to parental reports, the proband currently experiences up to 20 recurrent viral respiratory infections, often complicated by tracheobronchitis and significant oxygen desaturation. The proband was followed by a pulmonologist because of frequent lower respiratory tract infections and persistently low peripheral oxygen saturation, despite the absence of true hypoxemia.

At the genetic examination at 4 years and 3 months of age, anthropometric measurements were as follows: weight 16 kg (−0.47 SD), height 105 cm (−0.13 SD), and head circumference 49 cm (−0.95 SD). Physical examination revealed skin pallor without any dysmorphic craniofacial features. At the time of evaluation, the parents reported exercise-induced fatigue and persistently low oxygen saturation levels, emphasizing that the chronically decreased oxygen saturation remained their major source of concern.

Laboratory findings demonstrated a normal arterial PaO_2_ level, and no evidence of anemia and true hypoxemia was observed. Detailed laboratory results of the proband are presented in Table 1.

Hb electrophoresis showed decreased HbA (55.6%), with HbA_2_ and HbF within reference ranges, and the presence of an HbD fraction at 40.8%, which is normally absent, as shown in Figure 1.

Chest CT showed no pulmonary pathology or structural vascular anomalies, as shown in Figure 2a,b, whereas abdominal ultrasound and echocardiography also revealed no abnormalities.

Additionally, pulmonary perfusion scintigraphy was performed to assess regional pulmonary blood flow and to exclude ventilation–perfusion mismatch. The technetium-based radiopharmaceutical demonstrated uniform distribution throughout both lungs, with no focal perfusion defects. Images were obtained in the anterior and posterior projections 4 s after injection and in the frontal projection 300 s after injection, confirming homogeneous perfusion and the absence of regional abnormalities, as shown in Figure 3a–c.

Family history was notable in that the father had an isolated decrease in oxygen saturation (80–90%) in the absence of clinical signs of hypoxia.

### 2.2. Molecular Findings

WGS identified a heterozygous variant in the *HBB* gene, previously described as pathogenic: NM_000518.5:c.112T>C, p.(Trp38Arg). The variant was inherited from the father, as shown in Figure 4a,b. The identified nucleotide variant was not present in the Genome Aggregation Database (gnomAD v4.1.0, accessed 31 October 2025). This missense substitution affects a highly conserved residue of the β-globin chain: tryptophan at position β37(C3), a key contact point at the α1β2/α2β1 interfaces. Substitution with arginine alters polarity and disrupts these intersubunit interactions, destabilizing the quaternary structure and resulting in abnormal oxygen binding and release [7]. According to ACMG/AMP guidelines [11], the variant NM_000518.5:c.112T>C, p.(Trp38Arg), in the *HBB* gene meets criteria PM2, PP3, PM1, and PP4. Based on these criteria, this variant is classified as likely pathogenic. This variant has previously been reported in patients with pseudohypoxemia and is known as Hemoglobin Rothschild. The proband’s molecular and clinical findings are consistent with a hemoglobinopathy caused by an Hb D-region variant corresponding specifically to Hemoglobin Rothschild. In this context, the reference to an Hb D variant reflects electrophoretic co-migration into the Hb D zone rather than the presence of a classical Hb D structural variant. Recognition of this distinction is essential, as Hb Rothschild is a low-oxygen-affinity hemoglobin variant, accounting for the persistent pseudohypoxemia, the abnormal hemoglobin electrophoresis pattern, and the recurrent hypoxemic episodes. Based on this evidence, this variant is considered the cause of low SpO_2_ in our proband, rather than true hypoxia.

(a)Pedigree demonstrating autosomal dominant inheritance of the NM_000518.5:c.112T>C, (p.Trp38Arg) variant from the father.(b)WGS confirmation of the heterozygous variant in the proband and the father.

## 3. Discussion

Low-oxygen-affinity hemoglobin variants represent a diagnostically challenging group of hemoglobinopathies because they may produce persistently reduced SpO_2_ on pulse oximetry while maintaining normal arterial oxygenation. This discrepancy often results in diagnostic confusion and may lead to unnecessary cardiopulmonary evaluations or hospitalizations [12]. Hb Rothschild is one of the rarest variants in this group. Although the electrophoretic mobility of Hb Rothschild frequently overlaps with the Hb D region on hemoglobin electrophoresis, it differs fundamentally from classical Hb D variants in both molecular mechanism and functional impact: Hb Rothschild decreases hemoglobin–oxygen affinity rather than altering structural stability or causing clinically significant anemia.

Only one pediatric case of Hb Rothschild has been previously published, and that child was completely asymptomatic, with low SpO_2_ detected incidentally and no clinical complaints or functional limitations (summarized in Table 2) [10].

Comparison with previously reported carriers demonstrates a consistent biochemical profile across cases, including low peripheral oxygen saturation, normal PaO_2_, unremarkable cardiopulmonary imaging, and a substantial proportion of HbD on electrophoresis. Most reported cases involved adults in whom the variant was discovered incidentally. While dyspnea was occasionally noted, these symptoms were generally attributed to coexisting conditions rather than to the variant itself [5,6,7,8,9].

Our proband demonstrated the same characteristic biochemical pattern—elevated deoxyhemoglobin, reduced oxyhemoglobin, and a high HbD fraction—consistent with established structural consequences of the β37 (Trp38Arg) substitution. Familial segregation, with the father exhibiting isolated desaturation, further supports autosomal dominant inheritance with variable expressivity.

In contrast to previously described adults with largely asymptomatic courses, our case represents the first pediatric patient with detailed clinical characterization and clinically significant manifestations. Alternative explanations for chronic fatigue were effectively excluded through comprehensive cardiopulmonary, metabolic, and laboratory assessments, and trio-based WGS did not reveal additional pathogenic variants. Critically, exercise-induced fatigue occurred even in the absence of respiratory infections, indicating a chronic functional limitation rather than an illness-driven phenomenon.

Although low-oxygen-affinity hemoglobin variants generally facilitate oxygen unloading and are often clinically benign, the physiological context of early childhood differs substantially from that of adults. Children have higher metabolic demands and less mature compensatory mechanisms, potentially reducing their tolerance to chronic hypoxemia. The fatigue in our proband is most plausibly explained by reduced oxygen loading in the lungs rather than by impaired peripheral unloading. The right-shifted oxyhemoglobin dissociation curve enhances tissue oxygen release but simultaneously reduces hemoglobin’s ability to bind oxygen in pulmonary circulation. Consequently, a smaller proportion of hemoglobin becomes oxygenated even under normal alveolar partial pressures, producing persistently low SpO_2_ and elevated deoxyhemoglobin. In early childhood, this limited oxygen-loading capacity restricts physiological reserve during exertion, resulting in exercise-induced fatigue despite enhanced unloading.

Recurrent respiratory infections may exacerbate symptoms; however, the presence of fatigue between infections, combined with consistent biochemical findings and familial segregation, supports the Hb Rothschild variant as the primary contributor to exertional intolerance in this child.

Because this case highlights early-onset pseudohypoxemia, potential developmental implications must also be considered. In early childhood, even modest reductions in apparent oxygen saturation could theoretically affect growth, physical endurance, or neurocognitive development. Notably, our proband demonstrated age-appropriate growth and developmental milestones, and no cognitive concerns were identified during clinical evaluation.

Low-affinity hemoglobin variants are defined by an increased p50 value—the partial pressure of oxygen at which hemoglobin is 50% saturated—which serves as a functional marker of oxygen affinity: a high p50 reflects low affinity and enhanced oxygen release, whereas a low p50 reflects high affinity and impaired oxygen release [13]. Variants with low affinity, such as Hb Rothschild, Hb Kansas, and Hb Bassett, shift the oxygen dissociation curve to the right, allowing increased oxygen unloading into tissues; anemia, if present, is typically mild and does not require treatment [12,14,15,16,17,18].

Conversely, high-affinity hemoglobin variants stabilize the R-state and impair oxygen unloading, often causing secondary erythrocytosis and sometimes requiring intervention [19].

Previous work by Fomenko et al demonstrated that individuals with Hb Rothschild maintain normal exercise capacity despite low SpO_2_ readings, supporting the interpretation that pulse oximetry underestimates true oxygenation in these patients [6]. Although most reported adult cases confirm the benign nature of the variant, our case shows that clinically significant manifestations may emerge in early childhood, before compensatory mechanisms are fully developed.

In summary, Hb Rothschild is a rare low-affinity hemoglobin variant that produces chronically reduced peripheral oxygen saturation despite normal arterial oxygenation. Recognition of this pattern is critical, as reliance on pulse oximetry alone may lead to misdiagnosis and unnecessary interventions. Early genetic confirmation facilitates accurate diagnosis, appropriate management, and effective family counseling.

## 4. Conclusions

This case underscores the importance of recognizing low-oxygen-affinity hemoglobin variants as a cause of persistently reduced SpO_2_ despite normal arterial oxygenation. Unlike previously reported individuals with Hb Rothschild—including the only asymptomatic pediatric carrier described to date—our patient developed clinically significant symptoms in early childhood, demonstrating that Hb Rothschild is not always benign and may present with exercise intolerance, recurrent infections, and developmental concerns.

Clinicians should be aware that low SpO_2_ readings alone may lead to misinterpretation of hypoxemia and unnecessary cardiopulmonary investigations. Persistent desaturation with normal PaO_2_ should prompt consideration of hemoglobin variants, and early genetic testing supports accurate diagnosis, appropriate management, and targeted genetic counseling. Awareness of this entity among cardiologists, pulmonologists, anesthesiologists, and intensive care physicians is essential to prevent unwarranted interventions.

## Figures and Tables

**Figure 1 diagnostics-15-03181-f001:**
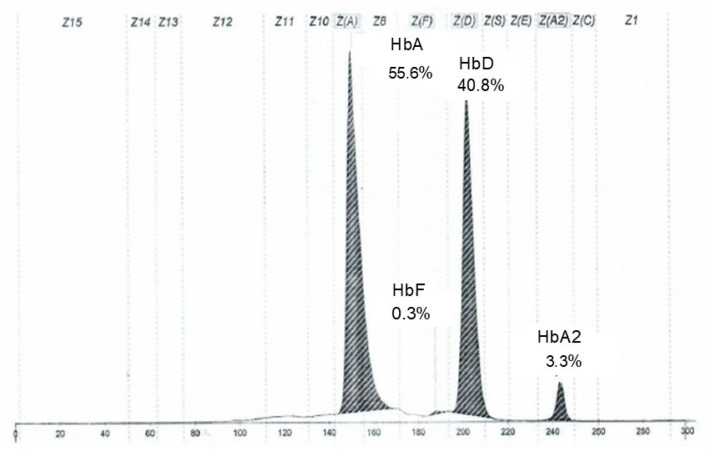
Electropherogram of the proband’s hemoglobin fractions obtained by capillary electrophoresis.

**Figure 2 diagnostics-15-03181-f002:**
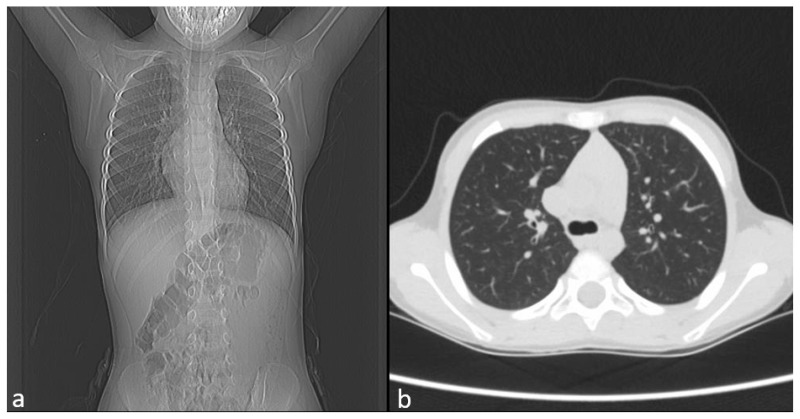
Image of CT. (**a**) Frontal view. (**b**) Axial view.

**Figure 3 diagnostics-15-03181-f003:**
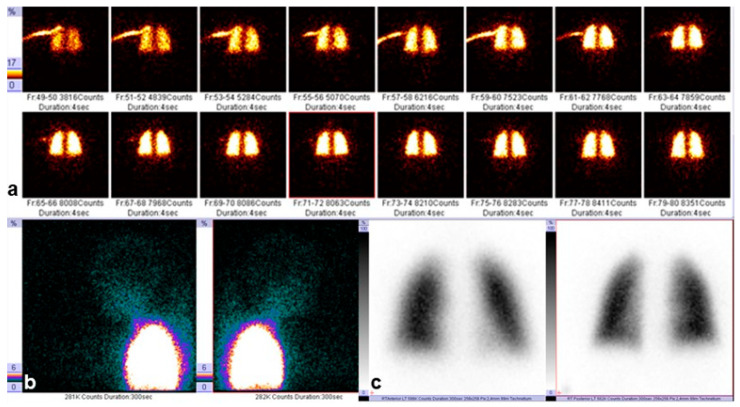
Pulmonary perfusion scintigraphy: (**a**) anterior view obtained 4 s after injection of the technetium-based radiopharmaceutical; (**b**) posterior view obtained 4 s after injection; (**c**) frontal view obtained 300 s after injection. The color scale indicates the intensity of radiotracer accumulation: areas of highest uptake are shown in white/yellow, moderate up-take in red, and low uptake in blue/green. A symmetric distribution of the radiotracer is observed.

**Figure 4 diagnostics-15-03181-f004:**
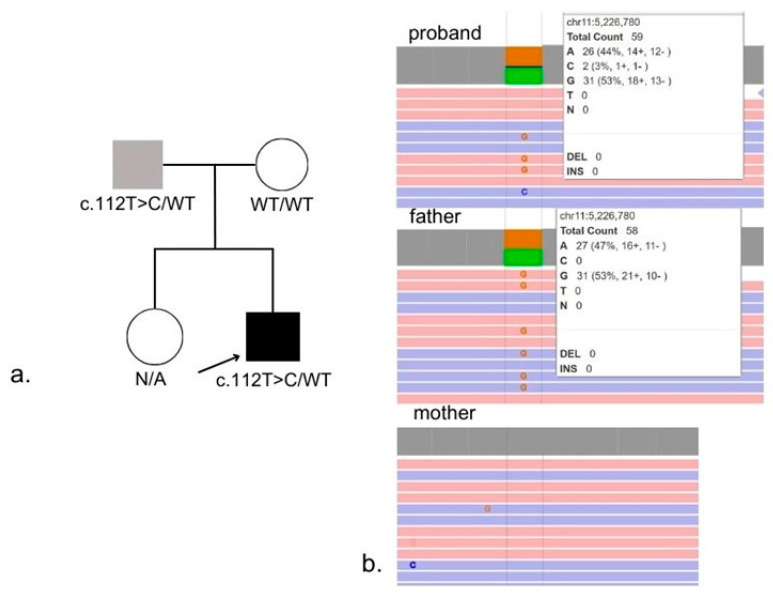
Family pedigree (**a**) and results of WGS (**b**).

**Table 1 diagnostics-15-03181-t001:** Blood test results.

Parameters	Reference	Results
Hemoglobin (g/dL)	11.5–14.0	11.9
MCV (fL)	77–87	79.2
MCH (pg)	24–30	28.1
MCHC (g/dL)	31–37	35.4
Reticulocytes (%)	N/A	1.67
WBC (10/L)	5.8–13.6	8.02
Platelets (10/L)	150–580	297
Urea (mmol/L) Creatinine (µmol/L)	2.5–6.4 27–62	within normal limits within normal limits
AST (U/L) ALT (U/L) Total bilirubin (µmol/L)	10–42 10–45 2–14	within normal limits within normal limits within normal limits

ALT—alanine aminotransferase; AST—aspartate aminotransferase; MCH—mean corpuscular hemoglobin; MCHC—mean corpuscular hemoglobin concentration; MCV—mean corpuscular volume; N/A—not available (reference range not provided by the laboratory); WBC—white blood cells.

**Table 2 diagnostics-15-03181-t002:** The comparison of the genetic and clinical features between our case and previously published cases with NM_000518.5:c.112T>C, p.(Trp38Arg).

P	Age	G	Family History	Symptoms	PH (Ref. 7.35–7.45)	PaO_2_ (Ref. 80–100 mmHg)	PaCO_2_ (Ref. 35–45 mmHg)	Bicarbonate (Ref. 22–26 mmol/L)	deoxyH (Ref. 2–5%)	oxyHb (Ref. 95–98%)	HbA (Ref. 94.5–97.3%)	Hb A2 (Ref. 2.5–3.5%)	Hb F	HbD (Not Found)	Pulse Oximetry, %
**Our proband**	4 y.o.	m	father carries the same variant and has chronically low SpO_2_ without symptoms	fatigue, episode of cyanosis during chest infection	7.41	95	38.9	22.8	**19.1**	**78.7**	**55.6**	3.3	0.3	**40.8**	**70–80 (ARI), 83**
**P1**, by Hladik A et al. [10]	8 y.o.	m	mother has the same variant and chronically low SpO_2_	no	N/A	**47**	N/A	N/A	N/A	N/A	N/A	N/A	N/A	N/A	**81**
**P2**, by Alli NA et al. [7]	27 y.o.	f	sibling affected; father also demonstrated low SpO_2_ readings	episode of cyanosis during chest infection in pregnancy	N/A	95	N/A	N/A	N/A	N/A	**58.2**	3.2	0.5	**38.1**	**81**
**P3**, by Li D et al. [8]	37 y.o.	m	multiple family members reported low SpO_2_; daughter carries the variant with elevated HbD fraction but remains asymptomatic	recurrent dyspnea; likely anxiety-related	7.39	94.2	43.6	25	N/A	N/A	**53.5**	3.3	0	**43.2**	**80**
**P4**, by Alexey Fomenko et al. [6]	61 y.o.	f	N/A	no	7.4	75 (ref. 71–104 mmHg)	41	26	N/A	N/A	N/A	N/A	N/A	N/A	**78**
**P5**, by González García LM et al. [9]	53 y.o.	m	Low SpO_2_ detected in sister and two nephews. Patient: former smoker (started at 13; quit 10 years ago; pack-years 15).	dyspnea on intense exertion	N/A	**41**	N/A	N/A	N/A	N/A	N/A	N/A	N/A	**35**	**76–77**
**P6**, by M. Bruns et al. [1]	53 y.o.	f	N/A	no	N/A	94	N/A	N/A	**12.9**	N/A	N/A	N/A	N/A	N/A	**83–84**

ARI—acute respiratory infection; G—gender; N/A—not available; P—patient; f—female; m—male; ref—reference rang. Values outside the reference range are shown in bold.

## Data Availability

The datasets used and/or analyzed during this study are available from the corresponding author upon reasonable request.

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
