# Peer review of "First Symptomatic Pediatric Case of Hb Rothschild (*HBB*: c.112T>C, p.Trp38Arg): Low-Oxygen-Affinity Hemoglobin Presenting with Persistent Pseudohypoxemia"

_diagnostics, 2025, doi:10.3390/diagnostics15243181_

Round 1
Reviewer 1 Report
Comments and Suggestions for Authors
The authors present a case report of a 4-year-old boy with Hb Rothschild. This case is clinically significant as it describes the first pediatric patient with this variant to present with symptomatic manifestations, including fatigue, growth issues, and recurrent respiratory infections. However, there are several things that need to be addressed before publication.
- In Figure 1, the electropherogram peaks are labeled as HgA, etc. The standard scientific abbreviation for Hemoglobin is Hb, not "Hg" (which represents Mercury). The authors must revise this figure to read HbA, HbD, etc., to maintain professional standards.
- Please standardize all references to "Figure" (e.g., Figure 2, Figure 4) in both the running text and the captions.
- As Diagnostics is an English-language journal, decimal points should be used instead of commas. This applies to the text body as well (e.g., "11,5-1,4" in Table 1).
- The discussion states that low-affinity variants facilitate increased oxygen unloading to tissues and are usually benign. However, the authors attribute the patient's severe symptoms (fatigue, muscle weakness) to the variant. If oxygen delivery to tissues is preserved or even enhanced (right shift), why does the patient suffer from exercise-induced fatigue? Is it possible the fatigue is related to the recurrent infections rather than tissue hypoxia per se? Please clarify.
- Please elaborate on the pathophysiology of the fatigue. Is it due to the reduced oxygen loading in the lungs (despite the right shift enhancing unloading)?
- While WGS is comprehensive, did the authors check for other variants in genes associated with respiratory health given the respiratory phenotype?
- Please provide the reference range for all testing parameters.
- Ensure all cells in all tables are filled or marked "N/A" consistently.
- Please elaborate more on the link between the genotype and the "symptomatic" infection phenotype in the discussion.
Author Response
Comments 1. In Figure 1, the electropherogram peaks are labeled as HgA, etc. The standard scientific abbreviation for Hemoglobin is Hb, not "Hg" (which represents Mercury). The authors must revise this figure to read HbA, HbD, etc., to maintain professional standards.
Response Thank you for pointing this now. We agree with this comment. The figure has been corrected: in Figure 1, the electropherogram peaks are now labeled HbA, HbD, etc.
Comments 2. Please standardize all references to "Figure" (e.g., Figure 2, Figure 4) in both the running text and the captions.
Response. Thank you for pointing this now. Everything has been brought into order. All references to “Figure” (e.g., Figure 2, Figure 4) have been standardized throughout both the running text and the figure captions to ensure consistency and clarity.
Comments 3. As Diagnostics is an English-language journal, decimal points should be used instead of commas. This applies to the text body as well (e.g., "11,5-1,4" in Table 1).
Response. Thank you for pointing this now. All values in Table 1 have been brought into full conformity with the journal’s English-language standards. Decimal points are now used instead of commas (e.g., “11.5–14.0”), ensuring consistency and correctness throughout.
Comments 4. The discussion states that low-affinity variants facilitate increased oxygen unloading to tissues and are usually benign. However, the authors attribute the patient's severe symptoms (fatigue, muscle weakness) to the variant. If oxygen delivery to tissues is preserved or even enhanced (right shift), why does the patient suffer from exercise-induced fatigue? Is it possible the fatigue is related to the recurrent infections rather than tissue hypoxia per se? Please clarify.
Response.
Thank you for this thoughtful comment. We agree that low–oxygen-affinity hemoglobin variants typically preserve or even enhance oxygen unloading and are often clinically benign. However, in our proband, alternative explanations for fatigue were effectively excluded: he underwent an extensive cardiopulmonary, metabolic, and laboratory evaluation, all of which were normal, and trio WGS revealed no additional pathogenic or likely pathogenic variants suggestive of neuromuscular, metabolic, or mitochondrial disease. Exercise-induced fatigue was observed even outside episodes of respiratory infection, indicating a chronic limitation rather than symptoms driven solely by intercurrent illness. Although a right-shifted oxyhemoglobin dissociation curve facilitates oxygen unloading, young children have higher metabolic demands and less mature compensatory mechanisms. In this context, the combination of persistently low SpO₂ and elevated deoxyhemoglobin in our proband likely reduces physiological reserve during exertion, contributing to exercise-induced fatigue despite facilitated unloading. Recurrent infections may further exacerbate symptoms, but the presence of fatigue between infections and the consistent biochemical and familial findings support the Hb Rothschild variant as the primary cause, with infections acting only as an additional stressor. Specifically, we have expanded the discussion of this issue in Discussion, page 14, lines 14–24,
and page 15, lines 1–7.
Comments 5. Please elaborate on the pathophysiology of the fatigue. Is it due to the reduced oxygen loading in the lungs (despite the right shift enhancing unloading)?
Response.
Thank you for this important comment. The fatigue observed in our proband is most plausibly explained by reduced oxygen loading in the lungs rather than impaired unloading in the tissues. Although low–oxygen-affinity hemoglobin variants shift the oxyhemoglobin dissociation curve to the right and facilitate oxygen release, they also reduce hemoglobin’s ability to bind oxygen in the pulmonary circulation. As a result, a smaller proportion of hemoglobin becomes oxygenated even at normal alveolar oxygen tensions. This leads to chronically low arterial oxygen saturation and elevated deoxyhemoglobin, both of which limit the total amount of oxygen delivered to tissues during exertion.
In young children—who have higher metabolic demands and less mature compensatory mechanisms—this reduced oxygen-loading capacity decreases physiological reserve. During physical activity, the limited rise in oxygenated hemoglobin becomes insufficient to meet increased metabolic needs, resulting in exercise-induced fatigue and reduced tolerance.
Thus, despite enhanced unloading (right shift), the primary issue is inadequate oxygen loading in the lungs, which explains the proband’s symptoms. Specifically, this explanation is now included in Discussion section, page 14, lines 23–24, and page 15, lines 1–7.
Comments 6. While WGS is comprehensive, did the authors check for other variants in genes associated with respiratory health given the respiratory phenotype?
Response. WGS was not the initial diagnostic step. Given the respiratory phenotype, targeted testing for Congenital Central Hypoventilation Syndrome was performed first, including analysis of PHOX2B polyalanine repeat expansions. These tests were negative, after which we proceeded to WGS to evaluate a broader set of genetic causes.
Comments 7. Please provide the reference range for all testing parameters.
Response. Thank you for pointing this now. All reference ranges have been provided for all testing parameters.
Comments 8. Ensure all cells in all tables are filled or marked "N/A" consistently.
Response. Thank you for pointing this now. All tables have been fully revised. Every cell is now either filled with the appropriate value or marked N/A consistently, as required.
Comments 9. Please elaborate more on the link between the genotype and the "symptomatic" infection phenotype in the discussion.
Response. Thank you for this valuable comment. We agree that the relationship between the genotype and the child’s infection-related symptoms requires clearer explanation. In the revised manuscript, we clarify that low–oxygen-affinity hemoglobin variants, such as Hb Rothschild, do not directly predispose patients to infections. However, persistently low SpO₂ and an increased proportion of deoxygenated hemoglobin may reduce physiological reserve during periods of illness, making otherwise mild respiratory infections more clinically apparent. In our proband, episodes of cyanosis and fatigue during infections likely reflect this reduced compensatory capacity rather than an increased susceptibility to infection itself. We have added this explanation to the Discussion on page 15 (lines 8–11 and onwards).
Reviewer 2 Report
Comments and Suggestions for Authors
Thank you to the authors for the opportunity to review this manuscript. The study reports a pediatric case of Hb Rothschild carrying a germline HBB c.112T>C variant. Although this mutation has been previously associated with disease in multiple reports, the distinct feature of this case lies in its early onset, which subsequently led to recurrent infections and related clinical manifestations. This highlights the importance of recognizing early disease presentation in childhood. Below are my specific comments.
- The accuracy and clarity of the language should be improved.
- In Table 1, many entries include commas. It is unclear what these represent—are they intended to be decimal points?
- The definitions of MCH and MCHC in the notation section of Table 1 are incorrect.
- The sentence “The chest CT, abdominal ultrasound, and echocardiography revealed no pulmonary pathology or structural vascular anomalies, as shown in Figures 2 (a, b)” (lines 105–106) requires revision. Figure 2 shows only a portion of the chest CT and does not include the other imaging studies.
- Please ensure consistent nomenclature for tables and figures. There are several instances of misuse in the manuscript, including lines 106, 108, and 125.
- When evaluating the pathogenicity of the variant using bioinformatic tools, the manuscript should specify which tools were used and summarize their prediction results.
- In lines 126–130, the authors state that the mutation disrupts inter-subunit interactions, resulting in abnormal oxygen binding and release. However, no evidence is provided to support these statements. Please cite appropriate references or include supporting molecular dynamics simulations or experimental data.
- Please consider restructuring Table 2 by switching the rows and columns, presenting the information horizontally, and using a standard three-line table format.
- It is important to clarify whether the proband carries any additional germline or de novo variants, as the reported symptoms cannot exclude the possibility of being associated with other compound mutations.
- Since the manuscript emphasizes the clinical significance of early onset, the Discussion section could be strengthened by briefly commenting on the potential effects of this mutation/disease on growth, development, and cognitive function compared with age-matched peers.
- Gene symbols should be defined upon first use, and all gene symbols must be italicized, including those appearing in the title.
Author Response
Comments 1. The accuracy and clarity of the language should be improved.
Response. Thank you for this comment. The manuscript has been carefully proofread, and the accuracy and clarity of the language have been improved throughout.
Comments 2. In Table 1, many entries include commas. It is unclear what these represent—are they intended to be decimal points?
Response. Thank you for the clarification. Yes, the commas represented decimal points, and we have corrected all such entries in Table 1.
Comments 3. The definitions of MCH and MCHC in the notation section of Table 1 are incorrect.
Response. Thank you for pointing this out. The definitions of MCH and MCHC in the notation section of Table 1 were indeed incorrect, and we have now corrected them.
Comments 4. The sentence «The chest CT, abdominal ultrasound, and echocardiography revealed no pulmonary pathology or structural vascular anomalies, as shown in Figures 2 (a, b)» (lines 105–106) requires revision. Figure 2 shows only a portion of the chest CT and does not include the other imaging studies.
Response. Thank you for this remark. We have revised the sentence accordingly. The correction has been implemented on page 7, lines 1–3, where the reference to Figure 2 now appropriately includes only the chest CT. Unfortunately, it is not possible for us to provide the echocardiography and abdominal ultrasound images.
Comments 5. Please ensure consistent nomenclature for tables and figures. There are several instances of misuse in the manuscript, including lines 106, 108, and 125.
Response. Thank you for this remark. All nomenclature for tables and figures has been reviewed and brought into full consistency throughout the manuscript.
Comments 6. When evaluating the pathogenicity of the variant using bioinformatic tools, the manuscript should specify which tools were used and summarize their prediction results.
Response. We have evaluated the pathogenicity of the variant according to the recommended criteria. The corresponding reference has been provided, and this information is now included in the manuscript on page 8, lines 14-15
Comments 7. In lines 126–130, the authors state that the mutation disrupts inter-subunit interactions, resulting in abnormal oxygen binding and release. However, no evidence is provided to support these statements. Please cite appropriate references or include supporting molecular dynamics simulations or experimental data.
Response. Thank you for this comment. A supporting reference has been added to substantiate the statement in lines 126–130.
Comments 8. Please consider restructuring Table 2 by switching the rows and columns, presenting the information horizontally, and using a standard three-line table format.
Response. Thank you for the suggestion. Table 2 has been revised and fully restructured: the rows and columns have been switched, the information is now presented horizontally, and a standard three-line table format has been applied.
Comments 9. It is important to clarify whether the proband carries any additional germline or de novo variants, as the reported symptoms cannot exclude the possibility of being associated with other compound mutations.
Response. Thank you for this important comment. Since the proband underwent WGS in a trio format, we were able to comprehensively assess all potential variants. Other detected variants were inherited from healthy parents and did not match the clinical phenotype. Therefore, no additional germline or de novo variants that could explain the pseudohypoxemia were identified. WGS thus allowed us to exclude other possible genetic causes.
Comments 10. Since the manuscript emphasizes the clinical significance of early onset, the Discussion section could be strengthened by briefly commenting on the potential effects of this mutation/disease on growth, development, and cognitive function compared with age-matched peers.
Response.
Thank you for this valuable comment. We have now expanded the Discussion to address the potential impact of early-onset hypoxemia on growth, development, and cognitive function. Specifically, we note that although, in theory, even modest reductions in arterial oxygen saturation during early childhood could influence growth trajectories, physical endurance, or neurocognitive outcomes, our proband demonstrated age-appropriate growth and developmental milestones, and no cognitive delays were observed. This addition has been incorporated into the Discussion on page 15, lines 12–16.
Comments 11. Gene symbols should be defined upon first use, and all gene symbols must be italicized, including those appearing in the title.
Response. Thank you for this observation. The manuscript has been carefully proofread, and all gene symbols are now defined upon first use and consistently italicized throughout the text, including in the title.